# Nutrients cause consolidation of soil carbon flux to small proportion of bacterial community

Bram W. Stone [1,2 ✉], Junhui Li [2], Benjamin J. Koch[2,3], Steven J. Blazewicz [4], Paul Dijkstra[2,3], Michaela Hayer[2], Kirsten S. Hofmockel [1,5], Xiao-Jun Allen Liu [6], Rebecca L. Mau[2,7], Ember M. Morrissey [8], Jennifer Pett-Ridge [4,9], Egbert Schwartz[2,3] & Bruce A. Hungate [2,3]

Nutrient amendment diminished bacterial functional diversity, consolidating carbon flow through fewer bacterial taxa. Here, we show strong differences in the bacterial taxa responsible for respiration from four ecosystems, indicating the potential for taxon-specific control over soil carbon cycling. Trends in functional diversity, defined as the richness of bacteria contributing to carbon flux and their equitability of carbon use, paralleled trends in taxonomic diversity although functional diversity was lower overall. Among genera common to all ecosystems, *Bradyrhizobium*, the Acidobacteria genus *RB41*, and *Streptomyces* together composed 45–57% of carbon flow through bacterial productivity and respiration. Bacteria that utilized the most carbon amendment (glucose) were also those that utilized the most native soil carbon, suggesting that the behavior of key soil taxa may influence carbon balance. Mapping carbon flow through different microbial taxa as demonstrated here is crucial in developing taxon-sensitive soil carbon models that may reduce the uncertainty in climate change projections.

[1] Earth and Biological Sciences Directorate, Pacific Northwest National Laboratory, Richland, WA, USA. [2] Center for Ecosystem Science and Society, Northern Arizona University, Flagstaff, AZ, USA. [3] Department of Biological Sciences, Northern Arizona University, Flagstaff, AZ, USA. [4] Physical and Life Sciences Directorate, Lawrence Livermore National Laboratory, Livermore, CA, USA. [5] Department of Agronomy, Iowa State University, Ames, IA, USA. [6] Institute for Environmental Genomics, Department of Microbiology and Plant Biology, University of Oklahoma, Norman, OK, USA. [7] Pathogen and Microbiome Institute, Northern Arizona University, Flagstaff, AZ, USA. [8] Division of Plant and Soil Sciences, West Virginia University, Morgantown, WV, USA. [9] Life and Environmental Sciences Department, University of California Merced, Merced, CA, USA. ✉email: Bram.Stone@nau.edu

Global climate projections depend on estimates of soil carbon accumulation and decomposition[1–3], processes driven by microorganisms[3–6]. Given the vast diversity of soil microorganisms, different microbial taxa may have individualistic effects on C fluxes in soil[7], yet testing this idea has been challenging. Soils hold over twice as much organic carbon (C) as terrestrial vegetation, and soil C turns over much more slowly. Soil microbial communities contain thousands of different heterotrophic microbial taxa that, together, influence soil C content, but the quantitative contributions of individual microbial taxa to the processes governing soil C accumulation and loss are not known. While some soil biogeochemical processes are physiologically specialized and dominated by a few phylogenetically specific groups, processes involved in heterotrophic decomposition are broadly distributed across the bacterial tree of life[8]. With many taxa contributing to the same process, the functional evenness of heterotrophic decomposition might be expected to be approximately equivalent to the evenness in abundance of heterotrophic decomposers, with each taxon contributing to decomposition in proportion to its abundance. As bacterial abundances are logarithmically distributed[9], we might expect that the contributions to soil C may be similarly distributed despite differences in ecosystem or bacterial community composition.

We used a combination of measurements and models to evaluate the contributions of individual bacterial taxa to heterotrophic growth and respiration in four soils along a climate gradient in northern Arizona. Taxon-specific growth rates were measured using quantitative stable isotope probing with [18]O-water (qSIP, see "Methods" section)[7,10] for soils collected from desert grassland (GL), Piñon-Juniper scrubland (PJ), Ponderosa Pine forest (PP), and mixed conifer forest (MC) sites, as described previously[11–13]. Mean annual temperature for all respective sites: 8.5, 7, 5.5, and 4 °C and mean annual precipitation: 230, 380, 660, and 790 mm[12]. To determine how taxon-specific contributions to growth and respiration varied with resource availability, measurements were conducted in the laboratory using unamended soil, soil with supplemental glucose, and soil with glucose plus a nitrogen source accessible to microbes, $[NH4]_2SO4$ (carbon +nitrogen). Isotopic signatures of specific 16S sequences were combined with 16S abundances from quantitative PCR to yield quantitative estimates of taxon-specific population size and growth.

per unit soil, as well as 16S copy number and genome size (as as per Li et al.[13]) to estimate taxon-specific cell size and carbon content (see "Methods" section). We estimated taxon-specific bacterial respiration as a function of taxon-specific growth rate and taxon-specific carbon use efficiency (CUE), using several parameterizations of the growth ~ CUE relationship (Supplementary Fig. 1, see "Methods" section). The relationship between microbial growth and efficiency is complex and difficult to identify based on existing literature[14]. Among models with different parameterizations, a unimodal relationship between growth rate and CUE was selected with the lowest AIC and further discussed in the methods (Table 1).

We compared this model to one without taxonomy-informed genome characteristics (16S content and genome size estimates) and without taxon-specific growth, in which individual bacterial taxa respired in direct proportion to their 16S abundance per unit soil. This comparison served to demonstrate the extent that 16S abundance data of the bacterial community alone can predict soil carbon flux. Across the four soils and nutrient amendment treatments, modeled respiration of individual bacterial taxa was summed over the bacterial assemblage and was compared with measured total soil respiration. When based on measured per-taxon growth rates, modeled bacterial respiration was positively related to total soil respiration ($R^2 = 0.83$, $p < 0.001$; Fig. 1a). In contrast, when estimated in proportion to a taxon's abundance alone, modeled bacterial respiration demonstrated a comparatively poor correlation ($R^2 = 0.02$, $p = 0.70$; Fig. 1b). Although our methods track the incorporation of [18]O-labeled water into bacterial DNA, and not carbon explicitly, these results indicate that growth of individual bacterial taxa measured through [18]O assimilation can be directly associated with the movement of C through the soil. For all but two soil and treatment combinations, modeled respiration was lower than measured respiration, likely in part owing to non-bacterial contributions to measured total respiration (which were not modeled). When we amended soils with carbon (C) and carbon with nitrogen (C + N) we found elevated soil respiration, patterns which were also observed with modeled bacterial respiration (Fig. 1a). Nutrient amendments also stimulated taxon-specific bacterial respiration ($F_{2,9} = 27.2$, $p < 0.001$) and productivity ($F_{2,9} = 6.96$, $p = 0.01$) leading to higher total C use in these treatments (Fig. 2). Generally, organisms that produced more biomass also respired more (Supplementary Fig. 2).

## Results and discussion

**Bacterial efficiency and respiration.** Taxon-specific productivity (µg C g soil$^{-1}$ week$^{-1}$) was modeled as a function of per-capita growth rate, taking into account relative abundance, 16S content

**Distribution and consolidation of bacterial carbon use.** Soils amended with nutrients had higher productivity and respiration; however, in these soils, carbon use was less evenly distributed across the bacterial community, especially in soils provided with carbon

---

### Table 1 Comparison of per-taxon carbon use efficiency functions.

| Relation to community CUE | CUE (growth) | $\Delta AIC_{co2}$ | $\Delta AIC_{cue}$ | $\Delta AIC_{combn}$ |
|---|---|---|---|---|
| Constrained | Unimodal$_{0.5}$ | 2.98 | 5.20 | 11.17 |
|  | Linear positive | 6.10 | 0 | 12.21 |
|  | Unimodal$_{0.05}$ | 0 | 14.39 | 14.39 |
|  | Linear negative | 2.85 | 19.87 | 25.56 |
|  | Exponential decline | 4.38 | 20.33 | 29.08 |
| Unconstrained | Exponential decline | 17.43 | 46.25 | 81.10 |
|  | Linear negative | 21.96 | 40.89 | 84.80 |
|  | Unimodal$_{0.05}$ | 19.58 | 47.72 | 86.89 |
|  | Linear positive | 23.56 | 50.77 | 97.89 |
|  | Unimodal$_{0.5}$ | 23.47 | 50.99 | 97.94 |

Akaike information criterion values expressed as the difference from the model with the lowest error ($\Delta AIC$) returned from regression models under different assumptions of per-taxon carbon use efficiency (CUE) as a function of per-taxon growth rate denoted by the CUE(growth) column. Per-taxon CUE estimates were calculated either constrained by the minimum and maximum observed community-level CUE values or bounded only by 0 and 0.85 (unconstrained). For all regression models, both terms were z-transformed. $\Delta AIC_{co2}$ indicates the fit of summed per-taxon respiration to measured respiration. $\Delta AIC_{cue}$ indicates the fit of summed relative abundance-weighted per-taxon CUE to community-level CUE. $\Delta AIC_{combn}$ indicates the sum of 2($\Delta AIC_{co2}$) and $\Delta AIC_{cue}$. Subscripts following unimodal function names indicate whether maximum per-taxon CUE was centered over a growth rate of 0.5 or the global median growth rate of 0.05 observed across all taxa.

---

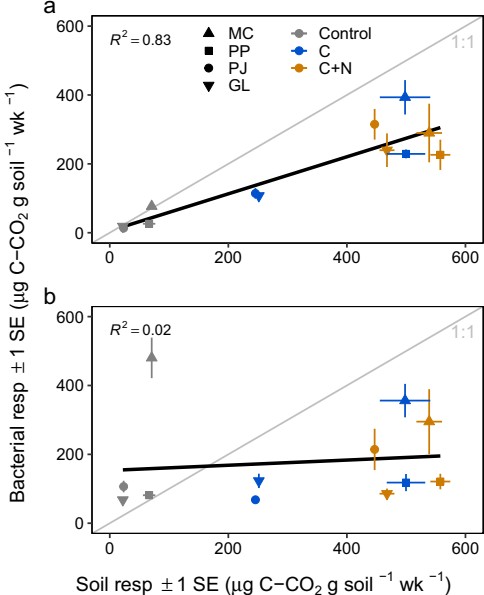

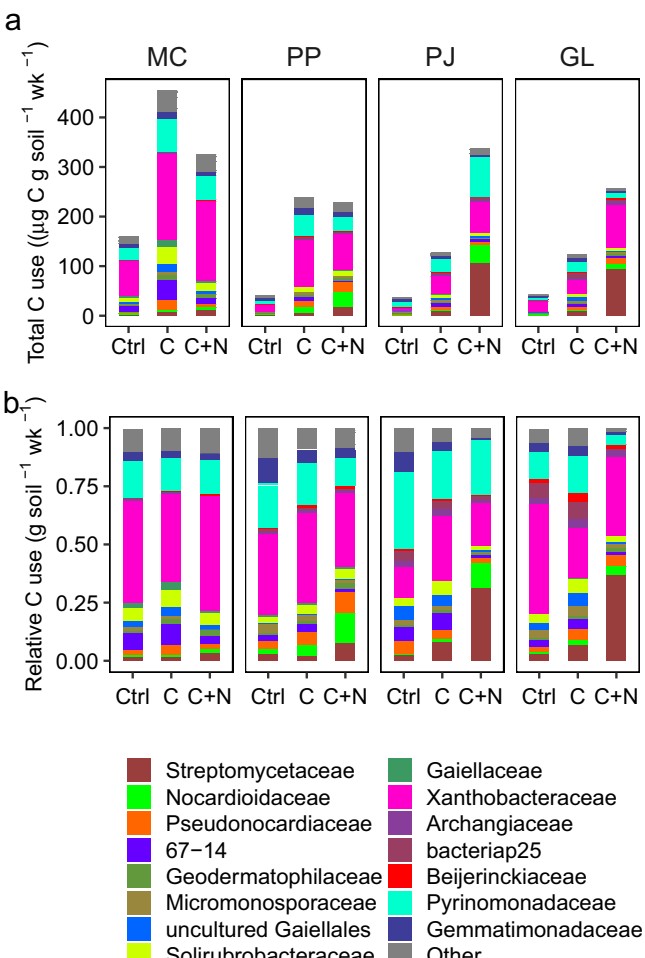

**Fig. 1 Fit of modeled respiration scaled from taxon-specific isotopic enrichment against community soil respiration, by mass of carbon (C) per g dry soil per week (wk). a** Bacterial respiration is estimated as the sum of modeled taxon-specific respiration and plotted against measured soil respiration. **b** Bacterial respiration is estimated from the community-level enrichment of all *16S* copies present in a sample (per g dry soil). Points show mean respiration values ± standard error (SE) across replicates ($n = 3$ experimental replicates) for each ecosystem (symbol MC mixed conifer forest, PP ponderosa pine forest, PJ piñon pine-juniper scrubland, GL desert grassland) and treatment (color control = no amendment, C = glucose, C + N = glucose and $[NH_4]_2SO_4$).

**Fig. 2 Absolute and relative carbon (C) use of bacterial families, per gram of dry soil per week (wk).** Values averaged across replicates for each ecosystem (MC mixed conifer forest, PP ponderosa pine forest, PJ piñon pine-juniper scrubland, GL desert grassland) by treatment (rows: Ctrl = no amendment, C = glucose only, C + N = glucose and $[NH_4]_2SO_4$) combination ($n = 3$ experimental replicates). Bar color represents bacterial family (15 shown, accounting for ≥75% of C use, remaining families designated as "Other"). **a** Total C use (C-$CO_2$ respired and MBC produced) from each bacterial family. **b** C use for each bacterial family, relativized by total C use.

and nitrogen. To compare the extent that taxonomic evenness equated to functional evenness (i.e., the extent that both shared similar abundance distributions), we calculated Pielou's evenness on the relative abundances of bacterial amplicon sequence variants (ASVs) as well as relativized growth and respiration estimates. Bacterial abundances were more evenly distributed than were estimates of bacterial productivity and respiration (Fig. 3a). Similarly, cumulative C use was strongly associated by treatment, with greater consolidation of carbon in C + N soils as shown by a lower proportion of the bacterial community responsible for a greater proportion of carbon flux (Fig. 3b).

Microbial community structure and function are thought to be linked[15,16], but most efforts to relate them rely on aggregate community function measurements correlated against summaries of composition, diversity, or interactions (e.g., Creamer et al.[17]). Interpretation of relative abundances across communities is a common exercise in contemporary studies of microbial ecology. Averaged across all ecosystems, 36 bacterial genera contributed to >50% of sequenced *16S* amplicons. Of genera common to all soils, only six were necessary to obtain >50% contributions to C cycling in control and C amended soils while only three were necessary to obtain >50% C cycling in C + N amended soils. *Bradyrhizobium* (Alphaproteobacteria, Family: Xanthobacteraceae), *RB41* (Acidobacteria, Family: Pyrinomonadaceae—Subgroup 4), and *Streptomyces* (Actinobacteria, Family: Streptomycetaceae) were common to all soils and treatments and in the C + N treated soils, these lineages accounted for the majority of C flux (Fig. 4a; Supplemental Table 1). These taxa also represent globally ubiquitous and abundant lineages as determined across the Earth Microbiome Project database[18].

Relative C use in the bacterial community was more consolidated within fewer lineages than the overall distribution of relative abundances might suggest. Averaged across all ecosystems and treatments, 75.7% of bacterial genera used less C than their relative abundance would otherwise predict. We assessed the relationship between relative C use and relative abundance in response to nutrient amendments using linear mixed modeling, accounting for random intercepts (and to limit pseudo-replication) across ecosystems and bacterial genera, and including an offset term to assess significant departure from the 1:1 line. In parallel with changing profiles of diversity, the relationship between taxon-specific bacterial C use and abundance was affected by treatment ($F_{2, 489.11} = 4.926$, $p = 0.008$). Specifically, we estimated that the slope of the relationship between relative C use and relative abundance was slightly but significantly higher than the 1:1 line in C + N amended soils, but not in control and C amended soils ($p = 0.02$) (Fig. 4b, c). Besides relative abundance, other potential influences on taxon-specific C

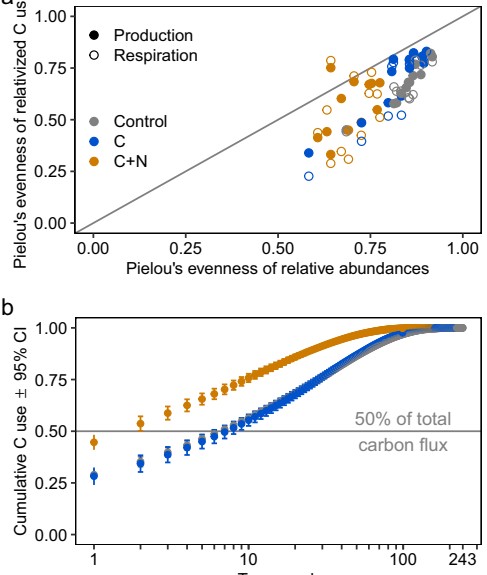

**Fig. 3 Change in bacterial taxonomic and functional evenness across soil nutrient amendments.** Color indicates soil treatment (Control = no amendment, C = glucose, C + N = glucose and [NH₄]₂SO₄). **a** Pielou's evenness of bacteria by relative abundance against Pielou's evenness by relativized carbon (C) use. Closed circles represent evenness of biomass production, open circles represent evenness respiration. **b** Cumulative contribution of bacteria to total relativized C use across soil amendment (n = 12 experimental replicates).

use estimates were per-capita growth rate and taxon-specific cell mass. Analysis of residual values from the linear mixed model found a significant positive relationship between per-capita growth rate and residual variation in C use ($F_{1, 583} = 14.3$, $p < 0.001$), whereas individual cell size (μg C) was not a significant driver, suggesting that the bacterial taxa that used more soil C in C + N soils did so because they grew and divided faster, not because they had larger cells, and that taxa that used less soil C in C + N soils grew more slowly than in other treatments. In addition, relative abundances likely reflect a mixture of both historical activity and activity as a result of the experimental conditions. A similar mixed model was therefore run using initial relative abundance, reflecting the historical activity of microbial taxa, as a predictor of C use during the incubation. Initial relative abundances were a significant model term ($F_{1, 375} = 83.3$, $p < 0.001$), suggesting that the historical activities of microorganisms can meaningfully influence the trajectories of microbial communities.

**Bacterial carbon use and abundance in response to resource stoichiometry.** In general, the relative contributions of individual bacteria to carbon use strongly resembled patterns of relative abundance, where the most abundant genera also utilized the largest proportion of C in the community (Fig. 4). However, while relative abundance was generally predictive of relative C use, it was difficult to predict any individual organism's contribution to C flux based on relative abundance alone, with differences between abundance and C use estimated to be an order of magnitude or more (Fig. 4b). One notable example was the genus *Sphingomonas* (Alphaproteobacteria) which had high *16S* abundance but contributed minimally to soil C flux (Fig. 4a). *Sphingomonas* could be distinguished from the top C using genera by a smaller cell mass estimate (31st percentile), a function

of genome length, which was lower than *Bradyrhizobium* (73rd percentile), *Streptomyces* (88th percentile), *RB41* (81st percentile) or the *Burkholderia-Caballeronia-Paraburkholderia* group (95th percentile), even though its growth rate was comparable.

In glucose-amended soils, the use of native soil C was closely correlated with the use of glucose across the bacterial community ($r = 0.96$, $p < 0.001$). In C and C + N treated soils, we performed ¹³C-glucose amendments in parallel to ¹⁸O-water conditions and used per-taxon ¹³C enrichment to estimate the amount of native (¹²C) and glucose carbon utilized across the bacterial community. Our results indicate that the organisms that utilized the most glucose were also those that utilized the most native soil carbon. Thus, organisms with the capacity to grow quickly in response to easily accessible carbon substrates are important to the cycling and turnover of existing soil C. To determine the extent that the C:N stoichiometry of labile resources may change microbial C use preferences, we used Levene's test on the variance in the relationship between ¹³C use and ¹²C use, where higher variance in response to nitrogen is indicative of shifts in the type of carbon preferred across bacterial genera. In C + N treated soils, there was significantly more variation around the trend line ($F_{22, 770} = 3.53$, $p < 0.001$, Levene's test; Fig. 4c), indicating that labile nitrogen addition may disrupt the balance between native soil carbon use and use of a labile carbon substrate.

Despite some differences between *16S* abundance and soil C use, across soils, differences in composition of the community significantly predicted the C use profiles ($r_M = 0.68$, $p < 0.001$, Mantel correlation). The four ecosystems differed in the amount of carbon used by different taxa ($R^2 = 0.69$, $p < 0.001$, PERMANOVA), patterns that mirrored differences in relative abundance ($R^2 = 0.69$, $p < 0.001$; Fig. 5). Similarly, bacterial communities changed in response to nutrient amendments, observed both with changes in relative abundance ($R^2 = 0.14$, $p = 0.03$) and C use ($R^2 = 0.13$, $p = 0.05$), though these differences were smaller than ecosystem-level separation of community composition and C use.

The strong ecosystem-specific clustering of community composition and C use (Fig. 5) is seemingly at odds with the strong treatment-specific patterns of cumulative C use (Fig. 3b), which suggests that there was a similar response to nutrient addition across all ecosystems regardless of which bacteria were responsible. However, relative abundance and C use were strongly linked (Fig. 4), and we observed that the most important contributors to bacterial C use were consistently represented by the same, abundant lineages across all ecosystems and treatments (Supplemental Table 1). Taken together, these results demonstrate that changing patterns in carbon use were driven by the consistent and abundant portion of the bacterial community which responded to the C + N amendment. Conversely, the importance of any individual lineage that occurred with low abundance towards soil C flux was difficult to determine. Rare taxa are thought to serve as a reservoir or seed bank of microbial function and diversity[19]. Although rare lineages drove ecosystem-specific patterns in community composition and C use due to the sensitivity of multivariate dissimilarity measures to their high diversity, differences in the composition of rare lineages were negligible, contributing minimally to soil C flux. These non-dominant organisms may be best described as part of the "interchangeable" biosphere, where apart from a few consistent taxa that dominate C flux, the identities of most rare taxa are negligible towards their contributions to C flux.

Generally, dominant lineages became more dominant—in terms of abundance as well as C use—in nutrient amended soils compared to native soil conditions, especially in the Actinobacteria. Consolidation of C use increased in C + N soils for most major phyla, where a proportionally smaller number of

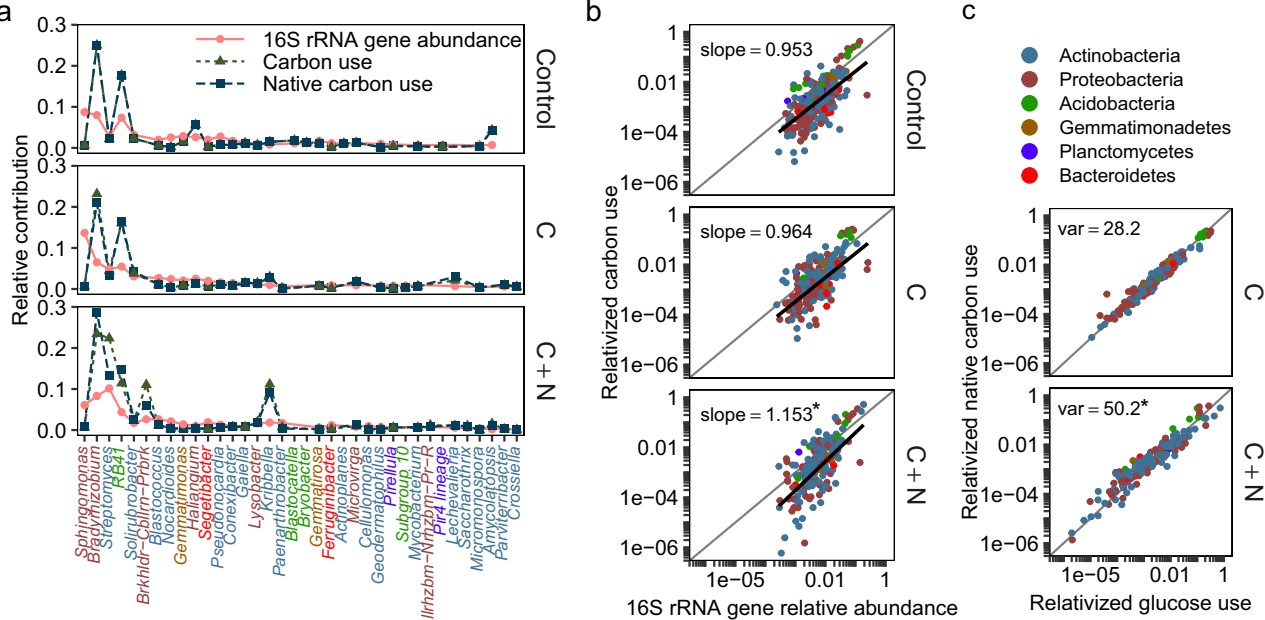

**Fig. 4 Comparison of relative abundance and relativized carbon (C) use of soil bacterial genera. Points show relative contributions from individual bacterial genera.** Values are averages across soil replicates from four ecosystems (mixed conifer forest, ponderosa pine forest, piñon pine-juniper scrubland, and desert grassland) and amended with either water (Control, labeled with $^{18}O$), glucose (C), or glucose and $[NH_4]_2SO_4$ (C + N) ($n = 3$ experimental replicates). **a** Comparison of the relative abundance and relativized C use of the top 36 most abundance genera. Colors correspond to bacterial phyla (six phyla accounted for >99% of C flux). Symbols correspond to the metric being compared for each taxon (relative abundance of *16S* rRNA gene amplicon sequences, relativized use of soil C, or relativized use of native soil C excluding added glucose). **b** Comparison of relative abundance and relativized C use across all genera. Trend lines show best fit from a linear mixed model accounting for differences between ecosystems and bacterial genera. Asterisks represent significant differences of slopes from the 1:1 line (two-sided unadjusted t-tests; C:Control $t_{287} = -0.60 \pm 0.06$ (std error), $p = 0.548$, effect-size $r = -0.0007$; C + N:Control $t_{489} = 2.42 \pm 0.06$, $p = 0.016$, effect-size $r = 0.012$). **c** Comparison of relativized glucose use and relativized native soil carbon use across all genera, with variance (var) around trend lines included. Asterisks represent significant differences in variance in C + N soils compared to C soils ($F_{22,770} = 3.53$, $p < 0.001$, Cohen's $d = 0.134$).

taxa was associated with a greater share of overall abundance and C flux in C and C + N treatments (Fig. 6). Several taxa within the Actinobacteria, mostly *Streptomyces* (Actinomycetaceae), *Arthrobacter* (Micrococcaceae), and *Kribbella* (Nocardioidaceae) spp., produced proportionally more *16S* rRNA gene copies than other Actinobacteria during the seven-day incubation in C and C + N soils (Supplementary Fig. 3). These taxa were also dominant producers of biomass and $CO_2$ even after correcting for *16S* rRNA gene copy number, cell mass, and growth rate (Supplementary Figs. 4–6). Across nearly all major bacterial phyla, the addition of labile nutrients tended to promote respiration of some lineages relative to others, increasing dominance, and demonstrating that the release of soil carbon as $CO_2$ can be concentrated in a few taxa (Figs. 3 and 6 and Supplementary Figs. 3, 4–6). These findings complement previous synthesis efforts, which have found key taxa are likely responsible for variability in carbon cycling[20]. Generally, microbial communities are more resilient to pulses, such as our C and C + N amendments, than longer disturbances (also known as press disturbances)[21]. It is possible that nutrient addition over longer periods would elicit a different response from abundant and rare bacteria as well as changes in overall soil productivity and respiration.

In conclusion, we identify the contributions of individual bacterial taxa to soil carbon flux through bacterial production and respiration in their native soil habitats, providing insight into the community dynamics that are missing in microbial carbon models[22,23]. Our model identified the growth of a few highly abundant bacterial lineages in response to labile nutrient additions, whose pre-existing high abundance in the community allowed them to assimilate ~50% of carbon consumed by or

available to bacteria. The well-known pattern of logarithmic bacterial frequency and abundance distributions, thus parallels the high importance of a relatively small subset of bacterial biodiversity in the carbon cycling of any given soil. Given that this pattern is universal in microbial communities[9], we expect that such inequality in carbon use is as well. 4 of the 20 most prolific contributors to soil respiration come from poorly understood bacterial groups, one from the Acidobacteria, a phylum often generalized as oligotrophic[24]. However, the abundance of individual bacterial taxa, alone, was not a sufficient predictor of soil C flux. Thus, the ability to measure in situ growth rates provided by techniques like qSIP has considerable potential to resolve the ecological roles of bacterial lineages that are difficult to culture, or whose functions would otherwise require extensive physiological assays. With regard to soil respiration modeling, we propose that because the majority of bacterial carbon flux could be accounted for by 3–6 common genera from ecosystems with different temperature and precipitation regimes, and that these genera were globally abundant and ubiquitous[18], it is worthwhile to determine both the global ubiquity and consistency in carbon process rates, as well as their determining traits, of such highly abundant bacteria in response to climate change. Doing so may reveal a core group of the soil microbial community that act as dominant carbon processors.

## Methods
**Sample collection and incubation**. Three replicates of soil samples were collected from the top 10 cm in of plant-free patches in four ecosystems along the C. Hart Merriam elevation gradient in Northern Arizona[25] beginning at high desert grassland (1760 m), and followed at higher elevations by piñon-pine juniper

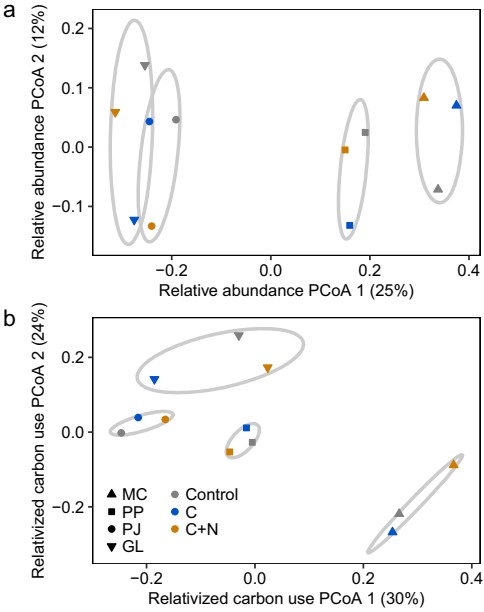

**Fig. 5 Composition of abundance and carbon use.** Ordinations generated by principal coordinates analysis (PCoA) of Bray–Curtis dissimilarities. Points represent centroids across replicates for each ecosystem (symbol MC mixed conifer forest, PP ponderosa pine forest, PJ piñon pine-juniper scrubland, GL desert grassland) and treatment (color control no amendment, C = carbon—glucose—only, C + N = carbon and nitrogen—$[NH_4]_2SO_4$) ($n = 3$ experimental replicates). Ellipses represent multivariate standard error ranges for ecosystem group position (95% confidence). Percentages along axes represent the percent of multivariate dispersion explained by each PCoA dimension. **a** Beta diversity of relativized abundances. **b** Beta diversity of relativized carbon use values (respiration plus biomass production, µg carbon per g soil per week).

woodland (2020 m), ponderosa pine forest (2344 m), and mixed conifer forest (2620 m). Soils were air-dried for 24 h at room temperature, homogenized, and passed through a 2 mm sieve before being stored at 4 °C for another 24 h. Soil incubations were performed on soils with mass of 20 g of dry soil for measurements of $CO_2$ and microbial biomass carbon (MBC), while 2 g of dry soil aliquots were incubated separately (but under equivalent conditions) for quantitative stable isotope probing (qSIP). We applied three treatments to these soils through the addition of water (up to 70% water-holding capacity): water alone (control), with glucose (C treatment; 1000 µg C g$^{-1}$ dry soil), or with glucose and nitrogen (C + N treatment; $[NH_4]_2SO_4$ at 100 µg N g$^{-1}$ dry soil). All samples for qSIP were incubated with $^{18}$O-enriched water (97 atom%) and matching controls necessary to calculate the change in $^{18}$O enrichment across the microbial community. We applied water at natural abundance (i.e., no $^{18}$O-enriched water) to the larger soil samples prepared for measurement of carbon flux. All soils were incubated in the dark for one week. Following incubation, soils were frozen at −80 °C for 1 week prior to DNA extraction.

**Soil, CO$_2$, and microbial biomass measurements.** We analyzed headspace gas of soils for $CO_2$ concentration and $\delta^{13}CO_2$ three times during the week-long incubation using a LI-Cor 6262 (LI-Cor Biosciences Inc. Lincoln, NE, USA) and a Picarro G2201 (Picarro Inc., Sunnyvale, CA, USA), respectively. Prior to incubation we analyzed soil MBC using the chloroform-fumigation extraction method on 10 g of soil. One sub-sample was immediately extracted with 25 ml of a 0.05 M $K_2SO_4$ solution, while a second sub-sample was first fumigated with chloroform (for 5 days), after which it was similarly extracted. Following $K_2SO_4$ addition, we agitated soils for 1 h, filtered the extract through a Whatman #3 filter paper, and dried the filtered solution (60 °C, 4 days). Salts with extracted C were ground and analyzed for total C using an elemental analyzer coupled to a mass spectrometer. MBC was calculated as the difference between the fumigated and immediately extracted samples' soil C using an extraction efficiency of 0.45 (as per Liu et al.[26]).

**Quantitative stable isotope probing.** We performed DNA extraction and *16S* amplicon sequencing on $^{18}$O-incubated qSIP soils[11–13]. The procedures targeted the V4 region of the *16S* gene as specified by the Earth Microbiome Project (EMP), http://www.earthmicrobiome.org) standard protocols[27,28]. We used PowerSoil DNA extraction kits following manufacture instructions to isolate DNA from soil

(MoBio laboratories, Carlsbad, CA, USA). We quantified extracted DNA using the Qubit dsDNA High-Sensitivity assay kit and a Qubit 2.0 Fluorometer (Invitrogen, Eugene, OR, USA). To quantify the degree of $^{18}$O isotope incorporation into bacterial DNA, we performed density fractionation and sequenced 15–18 fractions separately following methods modified from the canonical publication[7]. We added 1 µg of DNA to 2.6 mL of saturated CsCl solution in combination with a gradient buffer (200 mM Tris, 200 mM KCL, 2 mM EDTA) in a 3.3 mL OptiSeal ultra-centrifuge tube (Beckman Coulter, Fullerton, CA, USA). The solution was centrifuged to produce a gradient of increasingly labeled (heavier) DNA in an Optima Max bench top ultracentrifuge (Beckman Coulter, Brea, CA, USA) with a Beckman TLN-100 rotor (127,000 × g for 72 h) at 18 °C. We separated each sample from the continuous gradient into approximately 20 fractions (150 µL) using a modified fraction recovery system (Beckman Coulter). We then measured the density of each separate fraction with a Reichart AR200 digital refractometer (Reichert Analytical Instruments, Depew, NY, USA) and retained fractions with densities between 1.640 and 1.735 g cm$^{-3}$. We cleaned and purified DNA in these fractions using isopropanol precipitation, quantified DNA using the Quant-IT PicoGreen dsDNA assay (Invitrogen) and a BioTek Synergy HT plate reader (BioTek Instruments Inc., Winooski, VT, USA), and quantified bacterial *16S* gene copies using qPCR (primers: Supplementary Table 1) in triplicate. We used 8 µL reactions consisting of 0.2 mM of each primer, 0.01 U µL$^{-1}$ Phusion HotStart II Polymerase (Thermo Fisher Scientific, Waltham, MA), 1× Phusion HF buffer (Thermo Fisher Scientific), 3.0 mM MgCl$_2$, 6% glycerol, and 200 µL of dNTPs. We amplified DNA using a Bio-Rad CFX384 Touch real-time PCR detection system (Bio-Rad, Hercules, CA, USA) with the following cycling conditions: 95 °C at 1 min and 44 cycles of 95 °C (30 s), 64.5 °C (30 s), and 72 °C (1 min).

We sequenced the *16S* V4 region (primers: EMP standard 515F—806R; Supplementary Table 1) on an Illumina MiSeq (Illumina, Inc., San Diego, CA, USA). Sequences were amplified using the same reaction mix as qPCR amplification but cycling at 95 °C for 2 min followed by 15 cycles of 95 °C (30 s), 55 °C (30 s), and 60 °C (4 min). In addition to post-incubation soils, we extracted, amplified, and sequenced DNA of the bacterial community at the start of the incubation.

**Sequence processing and qSIP analysis.** The raw sequence data of forward and reverse reads (FASTQ) were processed within the QIIME 2 environment (release 2018.6)[29,30], denoising sequences with the available DADA2 pipeline[31]. We clustered the remaining sequences into amplicon sequence variants or ASVs (at 100% sequence identity) against the SILVA 132 database[32] using an open-reference Naïve Bayes feature classifier[33]. We removed global singletons and doubleton ASVs, non-bacterial lineages, and samples with less than 4000 sequence reads. Removal of global singletons and doubletons resulted in the removal of 2241 unique ASVs from the feature table yielding 115,647 out of 117,888 (a retention of 98% of all ASVs) as well as the loss of 4018 sequences leaving 37,765,678 (a retention >99% of all sequences). We combined taxonomic information and ASV sequence counts with per-fraction qPCR and density measurements using the phyloseq package (version 1.24.2), in R (version 3.5.1)[34]. Because high-throughput sequencing produces relativized measures of abundance, we converted ASV sequencing abundances in each fraction to the number of *16S* rRNA gene copies per g dry soil based on the known amount of dry soil added and the amount of DNA in each soil sample. All data and analytical code have been made publicly accessible[35].

To perform qSIP analysis and calculate per-capita growth rates of each ASV, we used our in-house qsip package (https://github.com/bramstone/qsip) based on previously published research[7,10]. Because rare and infrequent taxa are more likely to be lost in samples with poor sequencing depth with their absences affecting DNA density changes, we invoked a presence or absence-based filtering criteria on ASVs prior to calculation of per-capita growth rates. Within each ecosystem, we kept only ASVs that appeared in two of the three replicates of a treatment ($^{18}$O, C, and C + N) and at that appeared in at least five of the fractions within each of those two replicates. ASVs filtered out of one treatment were allowed to appear in another if they met the frequency threshold.

For all remaining ASVs (1081 representing less than 1% of all ASVs but 58% of all sequence reads), we calculated per-capita gross growth (i.e., cell division) rates observed in each replicate using an exponential growth model[10]. We applied these per-capita rates to the number of *16S* rRNA gene copies to estimate the production of new *16S* rRNA gene copies of each ASV per g dry soil per week using the following equation:

$$\frac{dN_i}{dt} = N_{i,t} - N_{i,t}e^{-g_i t}, \tag{1}$$

Where $N_{i,t}$ is the number of *16S* rRNA gene copies of taxon $i$ at time $t$ (here after 7 days) and $g_i$ represents the per-capita growth rate (calculated as a daily rate). See Supplementary Fig. 3 for results on the production of *16S* gene copies.

**Calculation of 16S rRNA gene copy numbers and cell mass.** In parallel to taxonomic assignment, we compared quality-filtered *16S* sequences against a database of 12,415 complete prokaryote genomes obtained from GenBank. From these genomes, we extracted data on *16S* rRNA gene copy number, total genome size, and *16S* gene sequence. We used BLAST to find matches against this database to the ASVs generated from QIIME 2 to make per-taxon assignments of *16S* rRNA

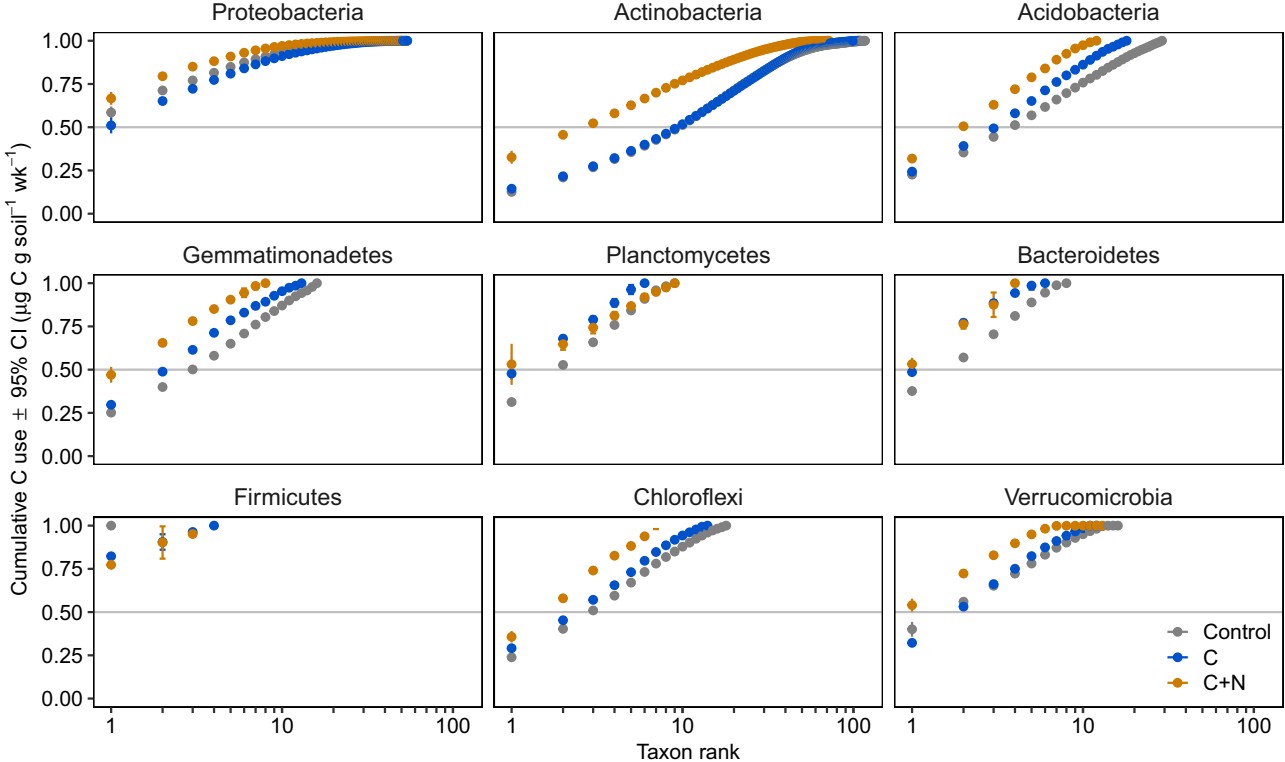

**Fig. 6 Change in functional evenness across soil nutrient amendments and major bacterial phyla.** Cumulative contribution of bacteria to total relativized carbon (C) use (the sum of μg C–CO₂ respired and μg MBC produced per taxon relativized by total C use across all taxa, both per g dry soil per week [wk]) across soil amendment. Points represent averages across soil replicates from four ecosystems (mixed conifer forest, ponderosa pine forest, piñon pine-juniper scrubland, and desert grassland) (n = 3 experimental replicates). Color indicates soil treatment (Control = no amendment, C = carbon—glucose—only, C + N = carbon and nitrogen—[NH₄]₂SO₄). Taxa were ranked by individual contribution to C use.

gene copy number and total genome size[13]. For ASVs that did not find an exact match, we assigned *16S* rRNA gene copy number values and genome sizes based on the median values observed in the most specific possible taxonomic rank. We estimated the mass of individual cells for each population using published allometric scaling relationships between genome length and cellular mass from West and Brown:[36]

$$\log_{10}(M_i) = \frac{\log_{10}(G_i) - 9.4}{0.24},\quad (2)$$

where $M_i$ indicates cellular mass (g) and $G_i$ indicates genome length (bp) for taxon *i*. We obtained this relationship by digitizing Fig. 4[36] using DataThief III and re-fitting the trend line in log–log space. We estimated that 20% of the cellular mass was carbon[37]. To validate this approach, cellular mass estimates and initial *16S* copy number measurements were used to estimate population-level biomass C values which were summed and compared to initial community-level MBC. We found that these values overestimated initial MBC by an order of magnitude. As such, cellular carbon mass was divided by 10 in our final calculations. We applied cellular mass and *16S* copy number estimates to the production of *16S* copies to estimate the production of biomass carbon for each taxon during the incubation period (*t*):

$$P_i = \frac{dN_i/dt}{C_i} \cdot M_i \cdot 0.2,\quad (3)$$

where $P_i$ indicates production of biomass carbon (μg C g dry soil⁻¹ week⁻¹) and $C_i$ indicates *16S* copy number per cell for taxon *i*. The 0.2 coefficient represents an estimate that 20% of cellular mass is composed of carbon.

**Efficiency and respiration modeling.** We estimated rates of respiration using qSIP-informed growth rates and community-level carbon use efficiency (CUE). CUE estimates were based on the incorporation of ¹⁸O-water into DNA as a measure of gross biomass production[38,39] and measured CO₂ in headspace gas from soil incubations. We calculated the production of ¹⁸O-labeled biomass carbon (¹⁸P) at the community-level for each sample by summing the products of per-taxon ¹⁸O enrichment (excess atom fraction, EAF) and relative abundance:

$$^{18}P = \sum_{i=1}^{n} (^{18}EAF_i \cdot y_i) \cdot DNA_0 \cdot f(MBC_0 \sim DNA_0),\quad (4)$$

where ¹⁸P indicates the gross production of ¹⁸O-labeled microbial biomass carbon

per gram of dry soil per week, ¹⁸EAF_i indicates the enrichment of DNA of taxon *i* and $y_i$ indicates its relative abundance, DNA₀ indicates the concentration of DNA per gram of dry soil prior to incubation, and MBC₀ indicates the microbial biomass carbon per gram of dry soil prior to incubation. Here, the MBC₀ ~ DNA₀ function indicates the linear relationship between MBC and DNA concentration. We used the output from Eq. 4 to calculate community CUE for each sample:

$$CUE = \frac{^{18}P}{(^{18}P + R)},\quad (5)$$

where *R* indicates the total CO₂ respired per gram dry soil per week.

We used the community CUE values from each sample (Eq. 5) to constrain/as upper and lower limits our estimates of per-taxon CUE. For a group of three replicates from a given ecosystem and treatment, we used the minimum and maximum observed community-level CUE values as the acceptable range of per-taxon CUE values. These constraints were used to control the shape of the function of per-taxon CUE and growth rate, though functions were modeled both with and without constraints (i.e., per-taxon CUE values were bounded only by 0 and 0.7). The range of community-level CUE values for each treatment were 0.18–0.53 for control soils, 0.04–0.13 for carbon amended soils and 0.03–0.08 for carbon and nitrogen amended soils and did not vary much between ecosystems. As a result of uncertainty in the literature about the relationship between growth rate and CUE[14], several different relationships were postulated to model per-taxon CUE as a function of per-taxon growth rate: linear increase, linear decrease, exponential decrease, unimodal with peak CUE at growth rate of 0.5, and unimodal with peak CUE at a growth rate of 0.05 (the median of all per-taxon growth rates in the data). Comparisons between functions were made by calculating AIC values from per-taxon respiration, summed, and regressing against measured respiration values. Likewise, for each function, we tested how well per-taxon CUE estimates reconstructed community-level CUE by weighting the CUE value of each taxon by its relative abundance, summing, and regressing against community-level CUE. To select the best per-taxon CUE function, AIC values from both scaling efforts were combined. To make AIC values comparable, all respiration and CUE terms were z-transformed prior to regression scaling. To reflect our priority of estimating per-taxon respiration, AIC values from the respiration scaling regression models were multiplied by two and summed with AIC values from CUE scaling such that AIC_Total = 2(AIC_Resp) + AIC_CUE. Across these comparisons, the best estimate of per-taxon CUE was the unimodal function of growth rate, constrained by

community-level CUE and peaking at growth rates of 0.5 (Table 1), such that:

$$\text{CUE}_i = -4(\text{CUE}_{E:T:range}) \cdot (g_i - 0.5)^2 + (\text{CUE}_{E:T:max}), \quad (6)$$

where $\text{CUE}_i$ indicates per-taxon CUE, $\text{CUE}_{E:T:max}$ indicates the maximum CUE values observed for a group of replicates within a given ecosystem and treatment (E:T). With this function, higher per-capita growth rate values were parameterized to produce higher CUE values initially and then decrease reflecting a growth-CUE tradeoff[14], here bound by the difference in maximum and minimum CUE values. We applied per-taxon CUE estimates from Eq. 6 to per-taxon growth rates to yield estimates of per-taxon respiration:

$$r_i = r_{g,i} + r_{m,i} = \left(\frac{g_i}{\text{CUE}_i} - g_i\right) + \left(\frac{g_i}{\text{CUE}_i} - g_i\right) \cdot \beta, \quad (7)$$

where $r_i$ indicates per-capita respiration for taxon $i$, $r_{g,i}$ indicates growth-related respiration, $r_{m,i}$ indicates maintenance-related respiration, and $\beta$ is a constant of 0.01 that represents the maintenance requirements as a proportion of total energy use[40]. We used these values of per-taxon, per-capita respiration rates to estimate per-taxon respiration per gram of dry soil per week:

$$R_i = P_i \cdot r_{g,i} + P_i \cdot r_{m,i}, \quad (8)$$

where $R_i$ indicates respiration of $CO_2$-C (μg C g dry soil$^{-1}$ week$^{-1}$) for taxon $i$.

In addition to per-taxon respiration estimates based on $^{18}O$ enrichment, we used another model for comparison. Here, respiration was calculated based on *16S* abundance alone:

$$R_i = N_i \cdot f(R \sim N + 0), \quad (9)$$

where $N_i$ indicates final *16S* abundance for taxon $i$, $R$ indicates microbial respiration of CO2-C (μg C g dry soil$^{-1}$ week$^{-1}$) and $N$ indicates total *16S* abundance at the end of the incubation. Here, the $R \sim N$ function indicates the linear relationship, with an intercept of 0, between $CO_2$ respiration and *16S* gene concentration across all samples.

**Diversity, compositional, and statistical analysis**. For patterns of evenness in bacterial carbon use and relative abundance, we used Pielou's evenness which is the quotient of Shannon's diversity and the observed richness. For each sample, we applied Pielou's evenness to bacterial abundances as well as bacterial carbon use (relativized to sum to one, in both cases).

We created a linear mixed model to test the relationship between the carbon use (the sum of biomass production and respiration) and relative abundance of bacterial genera from the dominant phyla, which accounted for >90% of all C flux. Here, we averaged carbon use and relative abundance for all replicates in a given ecosystem and treatment. We used the lme4 R package (version 1.1-20)[41] and obtained p-values using the Satterthwaite method in the lmerTest R package (version 3.1-0)[42]. To limit pseudo-replication, we accounted for differences in carbon use across ecosystems and due to bacterial Genus by implementing random intercepts. We selected for the optimal random and fixed components by dropping individual terms and comparing models with likelihood ratio tests, disregarding models that failed to converge. Our final model fit was:

$$\log_{10}(C_i) \sim \log_{10}(y_i)*T + (1|E) + (1|\text{Genus}), \quad (10)$$

where $C_i$ indicates the relativized carbon use for taxon $i$ (averaged across all three replicates in a given ecosystem and treatment), $y_i$ indicates the relative abundance of taxon $i$ (averaged across all three replicates), $T$ indicates soil treatment, and $E$ indicates ecosystem.

For differences in composition, we created species abundance tables using the traditional abundances, as well as measures of carbon use (growth and maintenance respiration) of each ASV in each sample. To account for differences in absolute abundances and flux rates between sites, we relativized all abundance tables. We summarized compositional differences using Bray–Curtis dissimilarities then identified multivariate centroids for all replicates in a site by treatment group. We tested the effect of site and nutrient amendment on the resulting group centroids using PERMANOVA tests implemented with the adonis function in the vegan package (version 2.5-3)[43]. We related compositional shifts in relative abundance to those in relativized growth and maintenance using Mantel tests with the mantel function in vegan.

To test for changes in the type of soil C preferred by microbial genera (either $^{13}C$-labeled glucose or $^{12}C$ soil carbon) in response to nitrogen addition, we used Levene's test with the car package (version 3.0-10)[44]. Specifically, we analyzed the relationship between $^{13}C$ use and $^{12}C$ use (both relativized) on bacterial genera across all replicates and in C and C + N treatments using a linear model. We then extracted model residuals and tested whether variance was significantly different across treatments by focusing on the interaction between individual replicates and treatment. This produced a significance test describing treatment-level differences in $^{13}C$–$^{12}C$ use.

**Reporting summary**. Further information on research design is available in the Nature Research Reporting Summary linked to this article.

## Data availability
Sequence data and sample metadata have been previously deposited in the NCBI Sequence Read Archive under the project number PRJNA521534. All other data have been made available at https://github.com/bramstone/bacterial-carbon-flux-qSIP (https://doi.org/10.5281/zenodo.4592585).

## Code availability
All statistical and modeling analyses have been made available at https://github.com/bramstone (https://doi.org/10.5281/zenodo.4592585).

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

## Acknowledgements

This research was supported by grants from the Department of Energy's Biological Systems Science Division Program in Genomic Science (Nos. DE-SC0016207 and DE-SCSC0020172), and by the National Science Foundation (No. DEB-1645596). Research conducted at Lawrence Livermore National Laboratory was supported by the U.S. Department of Energy Office of Science, via awards SCW1679 and SCW1590, conducted under the auspices of DOE Contract DE-AC52- 07NA27344. Research conducted at Pacific Northwest National Laboratory was supported by the U.S. Department of Energy Office of Science, via awards FWP 68907 and FWP 74475, conducted under the auspices of DOE Contract DE-AC05-76RL01830.

## Author contributions

B.W.S. analyzed data and wrote first draft of manuscript. J.L. performed bioinformatic analyses on prokaryote genomes. B.J.K., S.J.B., P.D., K.S.H., M.H., X.J.L., R.L.M., J.P.R., S.J.B., E.S. and B.A.H. contributed to project scope, experimental design, and interpreted findings. P.D., E.M.M. and B.A.H. contributed to organization and framing of manuscript. X.J.L., M.H. and R.L.M. collected and processed samples. M.H. and R.L.M. performed laboratory work and generated data. B.W.S. created supplemental Fig. 1. All authors contributed meaningfully to revisions.

## Competing interests

The authors declare no competing interests.
