## [Peer Review File · Nature Communications]

REVIEWERS' COMMENTS

Reviewer #1 (Remarks to the Author):

This is an interesting and important study that convincingly addressed the question of the role that different microbial communities play in soil C processing when soil is amended with available C and N inputs. I recommend the manuscript for publication, however, I have to say that I find it unfortunate that the authors decided not to address some of the issues that I have raised during the previous review of the manuscript. To me the manuscript's results became an exciting read starting from about page 6 - while the detailed description of the modeling processes in the Main part still seemed rather a distraction. I would still recommend either moving that to the Methods or provide a convincing justification of why that information is needed.

Just a couple of editorial suggestions:

l.49-55 I am not sure I see how this statement is related to the study. Here the CO₂ levels are not being changed and glucose additions of this study can not be directly translated into increased plant exudates. Maybe delete?

l.75-81. An extremely long sentence, most of which belongs to the methods.

l.94-97 I am not sure I see how this statement follows previous information.

2) l. 62. Is ammonium sulfate an organic N source? Or was there some other N source used?

3) l. 91-94. As Fig 1 suggests, these are not just modeled and observed respirations - these are bacterial modeled respiration vs. total soil respiration. This should be reflected in the writing, since at present it reads like modeled and measured respirations are the same thing.

4) l. 144-149. First of all, it is not clear the residuals from which models we are talking about here - because several different models were mentioned in the previous text. But, overall, it is unclear how and why associations between model residuals with per capita growth rate can inform relationships between C use and either growth rate or cell size? The link between them and its interpretation need to be explained in more detail here.

5) l.169-172. How was Levene's test conducted? It is not mentioned in the Methods. Also it is not clear what is the purpose of the Levene's test here - what is being compared to what? If it is used in a traditional manner here - to compare variances among the treatments or management practices, then something must be wrong with how the test was done, since degrees of freedom for the numerator cannot be 1.

Review comments for Stone *et al.* “Nutrients cause consolidation of soil carbon flux to small proportion of bacterial community”

Reviewer #1 (Remarks to the Author):

This is an interesting and important study that convincingly addressed the question of the role that different microbial communities play in soil C processing when soil is amended with available C and N inputs. I recommend the manuscript for publication, however, I have to say that I find it unfortunate that the authors decided not to address some of the issues that I have raised during the previous review of the manuscript. To me the manuscript's results became an exciting read starting from about page 6 - while the detailed description of the modeling processes in the Main part still seemed rather a distraction. I would still recommend either moving that to the Methods or provide a convincing justification of why that information is needed.

We thank the reviewer for their diligence throughout multiple revisions of our work. As suggested, we have moved the methodological details mentioned here (and detailed in the line-by-line comments) to the methods section. Additionally, we have removed descriptions of growth vs. maintenance related respiration in the main text as well as unnecessary descriptions of model comparisons in the succeeding paragraph. We agree that this makes the main text more accessible and focused on our key findings and believe that the reviewer should find these changes satisfactory.

Just a couple of editorial suggestions:

1.49-55 I am not sure I see how this statement is related to the study. Here the CO₂ levels are not being changed and glucose additions of this study can not be directly translated into increased plant exudates. Maybe delete?

We apologize for the confusion and have deleted the text as suggested.

1.75-81. An extremely long sentence, most of which belongs to the methods.

We agree that this sentence is very wordy and have removed it, as suggested. This description is already in the methods where it provides additional context to interested readers, and so it is not necessary here.

1.94-97 I am not sure I see how this statement follows previous information.

We agree that this statement could be clarified. We have amended it to better capture our intent which was that our methods track ¹⁸O-assimilation and not carbon assimilation directly. However, we could directly relate the growth of individual bacterial taxa (measured by ¹⁸O incorporation) to carbon flux (i.e., respiration) of the whole community. We have included phrasing to better communicate this:

Although our methods track the incorporation of ¹⁸O-labeled water into bacterial DNA, and not carbon explicitly, these results indicate that growth of individual bacterial taxa

measured through ¹⁸O- assimilation can be directly associated with the movement of C through the soil

2) l. 62. Is ammonium sulfate an organic N source? Or was there some other N source used?

We thank the reviewer for noticing this. In a strict sense, ammonium sulfate is not organic as there are no C atoms in ammonium sulfate. However, reduced N species are tightly cycled among microorganisms in the soil, who overwhelmingly drive the turnover of this N pool. As such, changes in the quantity and character of reduced N in soil may be essentially used as biological signatures. Microbial ecologists and soil ecologists commonly refer to reduced N species as “organic” as a result. However, our intent was to use more broadly recognized language, so we apologize for the confusion. We have amended our terminology and instead refer to ammonium sulfate as a “nitrogen source accessible to microbes.”

3) l. 91-94. As Fig 1 suggests, these are not just modeled and observed respirations - these are bacterial modeled respiration vs. total soil respiration. This should be reflected in the writing, since at present it reads like modeled and measured respirations are the same thing.

We thank the reviewer for this suggestion which should clarify our reporting. We have amended the use of “modeled respiration” to instead be “modeled bacterial respiration” and have amended “measured respiration” to instead be “measured total soil respiration” as indicated. We have made these edits in the sentence highlighted, and also in the preceding and succeeding sentences.

4) l. 144-149. First of all, it is not clear the residuals from which models we are talking about here - because several different models were mentioned in the previous text. But, overall, it is unclear how and why associations between model residuals with per capita growth rate can inform relationships between C use and either growth rate or cell size? The link between them and its interpretation need to be explained in more detail here.

We apologize for the confusion. Analyses of residual variation were used to follow up with the initial relationship between relative abundance and relative C use across bacterial taxa (i.e., the first mixed model described in lines 131-139). Different taxa were parameterized to have different cell masses as a function of genome size. Thus, a natural question to ask was “Since highly abundant organisms use more soil C, what other factors are important? Besides abundance alone, do larger cells or faster growing cells use more carbon?”

To clarify the purpose and specifics of these analyses, we have moved the reporting of all model residual analyses to be directly after the initial mixed model so that it is clear which model is being utilized. Further, we have explained why we conducted these analyses. We believe that these changes will allow readers to understand our motivations, methods, and results.

5) l.169-172. How was Levene's test conducted? It is not mentioned in the Methods. Also it is not clear what is the purpose of the Levene's test here - what is being compared to what? If it is used in a traditional manner here - to compare variances among the treatments or management practices, then something must be wrong with how the test was done, since degrees of freedom for the numerator cannot be 1.

We apologize that the Levene's test was omitted from the methods. Our intent was to use Levene's test in the manner described by the reviewer – to compare variances between the C and C+N treatments. However, we ran the Levene's test on data that were averaged across replicates – in essence, generating only one value in each of the two treatments (k) which yielded a numerator df of 1. This was a mistake on our part and we have amended our statistical code to compare the variance between treatments across all available replicates. We have updated the statistical reporting in the lines indicated (and in the methods, lines 608-614 of the revised manuscript). Testing across replicates, and focusing the interaction between replicates and treatment, the degrees of freedom in the test numerator are now 22. We sincerely thank the reviewer for catching this error.